# Impact of the COVID-19 Pandemic on Utilization of Inpatient Mental Health Services in Shanghai, China

**DOI:** 10.3390/healthcare10081402

**Published:** 2022-07-27

**Authors:** Hao Li, Xiaoli Chen, Jinhua Pan, Mengying Li, Meng Wang, Weibing Wang, Ying Wang

**Affiliations:** 1School of Public Health, Fudan University, Shanghai 200032, China; 20111020013@fudan.edu.cn (H.L.); 20211020192@fudan.edu.cn (X.C.); 19111020003@fudan.edu.cn (J.P.); 19211020107@fudan.edu.cn (M.L.); 20211020136@fudan.edu.cn (M.W.); 2Key Laboratory of Public Health Safety of Ministry of Education, Fudan University, Shanghai 200032, China; 3NHC Key Laboratory of Health Technology Assessment, Fudan University, Shanghai 200032, China

**Keywords:** mental disease, COVID-19, inpatient care

## Abstract

(1) Purpose: The ongoing COVID-19 pandemic has had an impact on mental health and the utilization of hospital-based inpatient mental health care worldwide. The aim of this study was to determine the impact of this pandemic on the utilization of this service in Shanghai by comparison with hospital-based health care records during the preceding 4 years. (2) Methods: The medical records were provided by the Shanghai Municipal Health Insurance Bureau. Diagnostic coding was based on International Classification of Diseases-10th revision (ICD-10), and inpatients with codes from F00 to F99 were examined. (3) Results: Inpatients were compared according to gender, age, pandemic stage, and type of mental disease. Utilization of psychiatric inpatient care in Shanghai during each of the four stages of the pandemic (1 January 2016 to 21 January 2020; 22 January 2020 to 9 February 2020; 10 February 2020 to 1 March 2020; 2 March 2020 to 31 July 2020) was analyzed. Before the lockdown, the utilization of psychiatric inpatient care had an overall upward trend; after the lockdown, the number of inpatients dropped sharply; as of 31 July 2020, it has not been restored. The utilization of this service for most types of mental disease declined rapidly during the pandemic; for vascular dementia (VAD, F01), it was relatively steady. The observed number of inpatient patients was about 51.07% lower than the predicted number in 2020. (4) Conclusions: The COVID-19 pandemic led to the implementation of prevention and control measures that reduced the utilization of psychiatric inpatient care in Shanghai. The use of inpatient services for categories F20–F29 had the greatest decline, and VAD (F01) had the smallest change during the pandemic. This service consequence of COVID-19 is apparent; to assure access to adequate service during a pandemic, health care professionals should pay close attention to changes in the utilization of different mental health services.

## 1. Background

The 2019 coronavirus disease (COVID-19) spread rapidly throughout the world since the end of January 2020, and many prevention and control measures were developed and implemented worldwide to reduce the most serious consequences. The World Health Organization (WHO) declared COVID-19 as a global public health emergency on 30 January 2020 [1,2]. As of 2 February 2021, China had 89,619 confirmed cases and 4646 confirmed deaths. To prevent serious public health problems and limit the spread of COVID-19, many regions implemented strict control measures, such as closing of schools and suspending all non-essential production and commercial activities. Undoubtedly, these measures affected daily public activities and the mental health of individuals [3]. A 2020 study showed that lockdown and social distancing measures during the COVID-19 pandemic led to loneliness, loss of income, reduced access to basic services, and decreased social support [4]. A study in Japan found that long-term home isolation during the COVID-19 pandemic led to lifestyle disruptions and an increased psychological burden, and even caused mental diseases in healthy people [5]. There is also evidence that the rapidly changing status of the pandemic led to widespread anxiety and distress [6]. While prevention and control measures can help to control a pandemic, they also affect the utilization of public health services. Since the beginning of the pandemic, China has restricted access to medical and health services and delayed some non-emergency medical care, resulting in reduced utilization of certain health services [7]. During this time, the pandemic had a serious impact on mental health [4,8]. However, many studies have shown that utilization of mental health services declined during the COVID-19 pandemic; there have been relatively few studies on the utilization of inpatient mental health services. For example, a study in France showed that during the first four weeks of the COVID-19 lockdown, psychiatric emergency consultations dropped by 54% compared to the same period in 2019 [9]. Two studies in Great Britain reported that the utilization of mental health services and hospital-based mental health care decreased sharply during the COVID-19 pandemic; compared to the same period in 2018 and 2019, the utilization of inpatient mental health care dropped by 20% [10,11]. To prioritize the hospitalization needs of COVID-19 patients, medical institutions have restructured and reorganized medical resources [12]. Some patients avoid hospitalization to reduce coronavirus infection [13]. However, other studies concluded the utilization of mental health services increased due to the increased psychological burden as the pandemic progressed [14,15]. Changes in the utilization of different mental health services during different stages of the COVID-19 pandemic are still unclear.

This study evaluated whether the implementation of prevention and control measures from 22 January 2020, to 31 July 2020 resulted in a shift in the numbers of inpatients at mental health institutions in Shanghai compared with that of the pre-pandemic period. We compared the number of observed inpatients with the forecasts for 2020 to assess the impact of the COVID-19 pandemic. Our general purpose was to examine the impact of the COVID-19 pandemic on the utilization of mental health services in an effort to prepare for the needs of medical services during the post-pandemic period.

## 2. Methods

### 2.1. Data Sources

This observational study reviewed the utilization of psychiatric inpatient care from 1 January 2016 to 31 July 2020 in Shanghai, China. All data were from the medical insurance records of workers and urban residents and were provided by the Shanghai Medical Insurance Bureau (SHIB), a government agency that has all local public hospitals under contract. The study protocol was reviewed and approved by the Ethics Committee of the School of Public Health, Fudan University (Permit Number: IRB#2020-11-0758). The variables examined included age, gender, main diagnosis, and length of hospital stay. All data were de-identified to protect patient privacy.

### 2.2. Statistical Analysis

The weekly numbers were analyzed using an interrupted time-series analysis to assess the effects of the COVID-19 pandemic by measuring outcome data. The period of the pandemic (1 January 2016 to 31 July 2020) was divided into four stages based on the epidemiological dynamics, containment policies, and health system policies in Shanghai. Stage 1 (1 January 2016 to 21 January 2020) was the pre-pandemic period, which ended with the establishment of the Joint Prevention and Control Mechanism. Stage 2 (22 January 2020 to 9 February 2020) was the peak-pandemic period, during which the strictest measures were implemented, including social distancing, enhanced contact tracing, testing, isolation, and quarantine, bans on public gatherings, and restrictions on movement, including stay-at-home orders and delays in returning to work and school. Stage 3 (10 February 2020 to 1 March 2020) was the post-peak pandemic period, which began with the postponement of elective healthcare services and ended with the restoration of these services. Stage 4 (2 March 2020 to 31 July 2020) was the post-pandemic period when Shanghai had the COVID-19 pandemic under control.

Monthly counts for each gender and age group were compared during the pandemic. The utilization of psychiatric inpatient care and different types of mental disorders between 2019 and 2020 were also compared. Daily scatter plots were used to describe changes in the numbers of inpatients during the different stages of the pandemic. Because of the small number of daily inpatients, analysis of inpatients with different specific diseases was not possible.

Normally distributed continuous variables were compared using the independent samples *t*-test, and non-normally distributed continuous variables using the Wilcoxon rank-sum test. Differences among age groups were compared using a one-way analysis of variance (ANOVA). The ARIMA model (moving average autoregressive model) was used for prediction in this study, which is the most common time series analysis and prediction model. Based on the daily cases of inpatients, according to the AIC (Akaike information criterion), many potential models were fitted and compared. All statistical analyses were performed using R software (version 4.0.3; R Project for Statistical Computing, Vienna, Austria) *p* values less than 0.05 were considered significant.

## 3. Results

### 3.1. Participants

Inpatients with codes from F00 to F99 were examined, and a total of 69,768 people took part in the study.

### 3.2. Descriptive Data

#### Characteristics of Inpatients

From 1 January 2016 to 31 July 2020, the SHIB recorded 69,768 visits to mental health institutions, with an increase over time (Table 1). There were 34,061 male inpatients (48.8%) and 35,707 female inpatients (51.2%). Analysis of inpatient age indicated that 13,569 (19.4%) were 44 years old or younger, 21,493 (30.8%) were 45 to 64 years old, 17,086 (24.5%) were 65 to 74 years old, and 17,620 (25.3%) were 75 years old or more. The length of hospital stays also declined significantly during this time (*p* < 0.001). Based on the ICD-10, most of these visits (66,029, 94.6%) were in four categories: organic, including symptomatic, mental disorders (F00–F09); schizophrenia, schizotypal, and delusional disorders (F20–F29); mood disorders (F30–F39); and neurotic, stress-related, and somatoform disorders (F40–F48).

### 3.3. Outcome Data

#### Overall Utilization of Psychiatric Inpatient Care during Different Stages of the Pandemic

During the first stage of the pandemic (1 January 2016 to 21 January 2020), the utilization of psychiatric inpatient care had an overall upward trend, with a relatively high level of utilization (Figure 1). During the second stage (22 January 2020 to 9 February 2020), presumably because of Shanghai’s joint prevention and control measures and the Chinese New Year holiday, which is a traditional festival in China during which most people will choose to go home, the number of inpatients dropped sharply (*p* < 0.001). During the third stage (10 February 2020 to 1 March 2020), when Shanghai gradually resumed normal medical and health services, the number of mental inpatients increased slightly (*p* = 0.299). However, the overall utilization of psychiatric inpatient care remained relatively low; as of 31 July 2020, the utilization of psychiatric inpatient care has not been restored (*p* < 0.001).

### 3.4. Main Results

#### Overall Utilization of Psychiatric Inpatient Care during Different Years

We compared the utilization of mental health institutions in terms of gender and age from January 2016 to July 2020 (Figure 2). Trends in the numbers of inpatients for men and women are basically the same during this period. The utilization of psychiatric inpatient care varied among age groups. Inpatients aged 45 to 64 years had the greatest utilization, and those aged 44 years or less had the least utilization (*p* < 0.001). Utilization of psychiatric inpatient care were compared for different conditions during different years and different stages of the pandemic.

From 2016 to 2019, the utilization of psychiatric inpatient care for six major types of mental illnesses generally had an upward trend (Figure 3). However, during the COVID-19 pandemic, the utilization of these services dropped significantly compared with the same period (January to July) in previous years (all *p* < 0.001). The average number of inpatients with ICD-10 codes of F00–F09 was about 2627 from 2016 to 2019, and there were only 2563 inpatients during the same period, 2.48% greater than in the same period of 2020.

The ten most common diagnostic codes of mental health inpatients varied from 2016 to 2020. From January to July of each year from 2016 to 2019, the overall utilization of psychiatric inpatient care increased, but there was an overall decrease during the same months in 2020, The numbers of inpatients during the four stages of the pandemic were 10.9%, 14.0%, 23.8%, and 29.7% lower than during the corresponding periods from 2016 to 2019. However, during the pandemic, the number of inpatients with vascular dementia (VAD) (F01) continued to increase, the number with unspecified dementia (F03) was greater than during the same period in 2016 and 2017, the number with somatoform disorders (F45) was greater than during the same period in 2016, and the number with recurrent depressive disorder (F33) was similar to that in 2018 (Figure 4).

From 1 January to 31 July 2020, similar to the same period in 2019, the two major disease codes were F00–F09 and F40–F48. Moreover, the utilization of psychiatric inpatient care during all periods of 2020 was less than during the same periods in previous years. The two major diseases (code F20–F29 and F30–F39) were more common in January 2020 than during the same period in 2019, but these numbers decreased rapidly in the subsequent months of 2020. In contrast, there were variations in the utilization of psychiatric inpatient care for codes F10–F19; in January, May and July of 2020, the number of inpatients with these diagnostic codes was greater than during the same periods in 2019, but this number was lower in the other months of 2020 (February, March, April, and June) than in 2019. During the pandemic, the number of inpatients with five major disease codes decreased sharply to their lowest points in February 2020. In particular, compared with January 2020, patients with code F00–F09 declined by 8.8%, patients with code F10–F19 declined by 22.4%, patients with code F20–F29 declined by 26.7%, patients with code F30–F39 declined by 20.9%, and patients with code F40–F48 declined by 12.2%. The overall utilization of psychiatric inpatient care increased during the post-peak pandemic stage. The number of inpatients with code F40–F48 had the greatest increase among these five types of mental illnesses relative to February 2020, and the number of inpatients with this code increased by 20.61% in April 2020 relative to February 2020. In contrast, the number of inpatients with the code F00–F09 had the smallest increase relative to February, and the number of inpatients with this code only increased by 8.4% during May (Figure 5).

The 10 most common disease codes during the pandemic accounted for 85.2% of all mental health inpatients. During January and February 2020, due to the pandemic, the Chinese New Year, and the implementation of preventive and control measures, the utilization of psychiatric inpatient care declined to their lowest levels for all 10 of these mental illnesses. Compared with January 2019, there were more inpatients with codes of F20, F29, F31, F41, and F79 in January 2020; however, these numbers were lower in all other months of 2020 than 2019. The numbers of inpatients with codes F06, F07, and F45 during 2020 were lower than during the same periods in 2019. The number of inpatients with code F03 was variable, as the utilization of psychiatric inpatient care during June 2020 was slightly higher than during June 2019 but the number during 2020 overall was less than during 2019 overall. The number of inpatients with code F01 was lowest in February 2020 but was slightly higher in 2020 than during the corresponding periods in 2019 (Figure 6). A comparison was made between observed inpatients and predicted inpatients from January to July 2020.

Figure 7 shows the trends in the utilization of psychiatric inpatient care from January to July 2020. The observed number of inpatient patients was significantly lower than the predicted number in 2020, and it was about 51.07% lower than the predicted number. During the COVID-19 period, average monthly visits dropped by approximately 1068 (*p* < 0.001) (Table 2).

## 4. Discussion

This study investigated the utilization of psychiatric inpatient care in Shanghai, China during the COVID-19 pandemic, and analyzed the numbers of inpatients according to gender, age, and specific mental disease categories. This result is consistent with other studies that reported decreased utilization of mental health services in some European countries during the COVID-19 pandemic. In the United Kingdom [10], the number of mental health inpatients declined significantly during the pandemic. In Germany [16], the COVID-19 outbreak led to an overall reduction of admissions to inpatient treatment. In Spain [17], the study showed an apparent decrease in the number of psychiatric emergency admissions during the lockdown due to the COVID-19 pandemic. In a Norwegian study [18], compared with the same period in 2019, the number of individuals seeking inpatient services for mental health declined from 16 to 7. Asian countries such as South Korea, Japan, and Australia, and the United States also found that the utilization of mental health services declined during the COVID-19 period [19,20,21,22]. This phenomenon may be the result of strict isolation measures and medical restrictions, which delayed patient admission plans. The study found that during the COVID-19 pandemic, staying at home was beneficial to patients because they could be supported by their families and avoid the stress of the work environment, to a certain extent reducing their demand for health services [23]. However, the reduction in the utilization of health services may lead to worsening patient conditions. The WHO report on mental health clearly stated that the COVID-19 pandemic had a serious impact on mental health services throughout the world in 2020 [24]. Although many studies showed that the utilization of mental health services declined during the pandemic, it is possible that many countries will experience a surge in the need for these services during their post-pandemic period [20].

Our analysis of the proportions of inpatients with different types of mental diseases indicated obvious changes during the pandemic. In particular, we found that from January to July of 2020 and during the same periods of 2016 to 2019, mental health patients with six disease codes accounted for more than 98% of inpatients. The most common disease codes were for symptomatic, mental disorders (F00–F09); mental and behavioral disorders due to psychoactive substance use (F10–F19); schizophrenia, schizotypal, and delusional disorders (F20–F29); mood disorders (F30–F39); neurotic, stress-related and somatoform disorders (F40–F48); and mental retardation (F70–F79). During Chinese official holidays, almost all hospitals do not accept patients except for emergency departments. In addition, compared with other holidays such as Tomb Sweeping Day, Labor Day, Dragon Boat Festival, the Chinese New Year holiday is longer. Meanwhile, influenced by traditional Chinese culture, Chinese people attach great importance to the family reunion during the Spring Festival. Therefore, the Spring Festival is an important confounding factor affecting the utilization of health services for patients. At the same time, the pandemic was at its peak and many cities were under strict lockdown [25], most hospitals in China provided mental health services to patients using remote medical management procedures [26], confinement measures, social isolation, and accessibility constraints, and there was a decline in the use of inpatient services between these group of patients [27]. The number of inpatients with codes F00–F09 and F10–F19 in 2020 was greater than during the same period in 2016 and 2017; however, this number was lower in 2020 than during the same periods of 2018 and 2019. Dementia is a neurodegenerative disease classified as F00–F09 whose patients often experience memory or cognitive problems and difficulties in behavior and daily life, and therefore often need more help to maintain normal lives. Social restrictions and containment policies during the pandemic had a negative impact on these patients. Reduced exercise and lack of social activities with relatives may aggravate the dementia of these patients and increase their need for mental health services [28]. During the COVID-19 pandemic, people’s consumption of alcohol increased, which may have led to the damage of mental health and increased their need for mental health services [16].

In this study, we found the observed number of mental health inpatients decreased significantly compared to the predicted number. The interrupted time-series analysis also observed the same changes in the recent trend. To prevent the spread of COVID-19, most countries adopted lockdown policies, such as restricting hospitalization and the movement of people among different regions and expanding social distance [20,29,30]. Many studies found significant reductions in mental health-related emergency department presentations and the number of beds and visits to psychiatric wards [20,22,31]. However, the utilization of mental health services varied by mental illness during the pandemic. On the one hand, some medication-based mental patients, such as those with bipolar disorder and anxiety, can obtain treatment through remote counseling as a supplement to hospital-based health services [32]. Therefore, the number of the above inpatients decreased significantly. On the other hand, the current findings showed the impact of the pandemic on the utilization of mental health services for VAD (F01) was much smaller compared to other mental illnesses. The number of inpatients with this diagnosis dropped sharply in February relative to January but was greater than during the same period in 2019. Vascular dementia (VAD) refers to a chronic condition caused by cerebrovascular disease, for example as a result of multiple prior infarctions in the brain. VAD is one of the most common types of dementia; affected by the pandemic, dementia patients are prone to symptoms of apathy, agitation, and anxiety, leading to a deterioration of their condition [33,34]. This explains why the number of inpatients with VAD remained relatively unchanged during the pandemic.

This study analyzed data on the utilization of mental health services in public hospitals in Shanghai from 1 January 2016 to 31 July 2020. The strengths of this study are that we covered a relatively long time span and the sample size was large. Further, we considered all mental diseases with ICD-10 codes of F00–F99 and separately analyzed the use of medical care for different diseases. This study also had some limitations. First, this study was based in Shanghai, so ITS generalizability to regions outside China may be questionable, although Shanghai may be considered representative of other large cities in China. In subsequent studies, we plan to cooperate with researchers in other regions of China and other countries to examine the impact of the COVID-19 pandemic on the utilization of inpatient services by mental health patients. There may have been some selection bias because we only obtained data on inpatients registered in the SHIB and we did not consider outpatients, emergency patients, and transient individuals. In addition, our study did not examine any clinical or detailed sociodemographic information, such as patient living situations, psychopathological aspects, substance use, etc. Nonetheless, the SHIB covers 95% of the residents in Shanghai, and we believe they are representative of the overall population.

## 5. Conclusions

Our major finding is that the COVID-19 pandemic and the resulting disease control policies reduced the utilization of psychiatric inpatient care and the number of daily inpatients in Shanghai. Public health professionals need to pay more attention to the negative impact of the pandemic on mental health. We assessed the impact of various policies on the receipt of inpatient mental health services and identified clear changes in the demand for these health services. The need for inpatient services for patients with mental health diseases has a relatively large elasticity, so it is necessary to ensure the availability and effectiveness of patient services during the post-pandemic period. The quantity and types of health services should be considered when attempting to implement the most reasonable allocation of medical and healthcare resources during the different stages of a pandemic.

## Figures and Tables

**Figure 1 healthcare-10-01402-f001:**
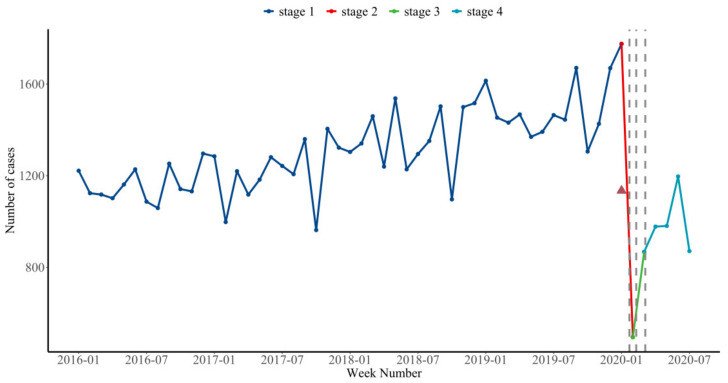
Interrupted time-series of inpatients with mental health between January 2016 and July 2020 in Shanghai. Legend: The dark blue line = pre-pandemic period, stage1 (1 January 2016 to 21 January 2020); the red line = peak-pandemic period, stage2 (22 January 2020 to 9 February 2020); the green line = the post-peak pandemic period, stage3(10 February 2020 to 1 March 2020); the light blue line = the post-pandemic period, stage4 (2 March 2020 to 31 July 2020). The red triangle is the day of Chinese New Year. The dashed vertical line represents policy nodes. Vertical lines represent the changes in prevention and control measures in Shanghai.

**Figure 2 healthcare-10-01402-f002:**
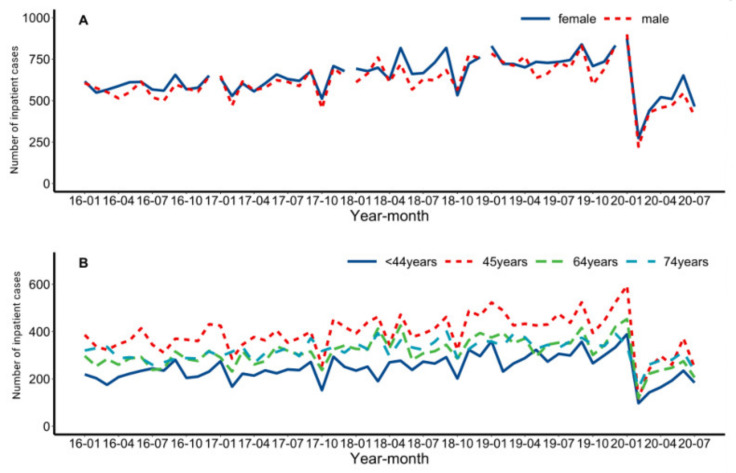
Monthly utilization of healthcare services from January 2016 to July 2020. Legend: Monthly numbers of inpatient visits in men and women (**A**), and in subjects aged ≤44, 45–64, 65–74 and ≥75 years (**B**).

**Figure 3 healthcare-10-01402-f003:**
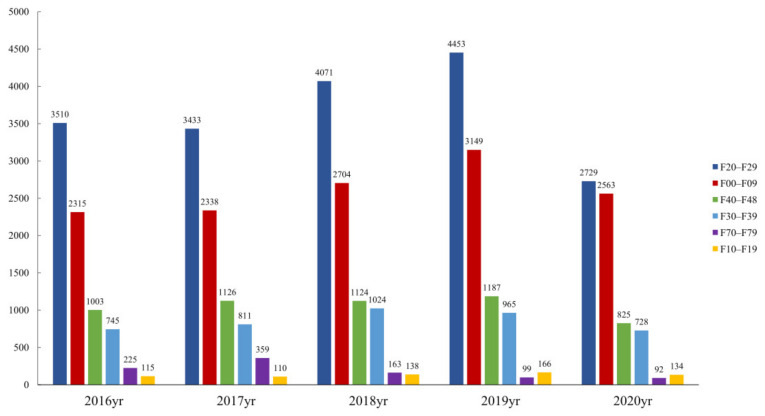
Utilization of the six categories of mental health services at different years. Legend: From January to July 2020 and the same period from 2016 to 2019, the utilization of the six categories of mental health services, F01–F09, F10–F19, F20–F29, F30–F39, F40–F48, and F70–F79. Note: F00–F09, organic (including symptomatic) mental disorders; F10–F19, psychoactive substances mental disorders; F20–F29, schizophrenic disorders; F30–F39, mood disorders; F40–F48, neurotic, stress-related and somatic disorders; F70–F79, neurodevelopmental retardation.

**Figure 4 healthcare-10-01402-f004:**
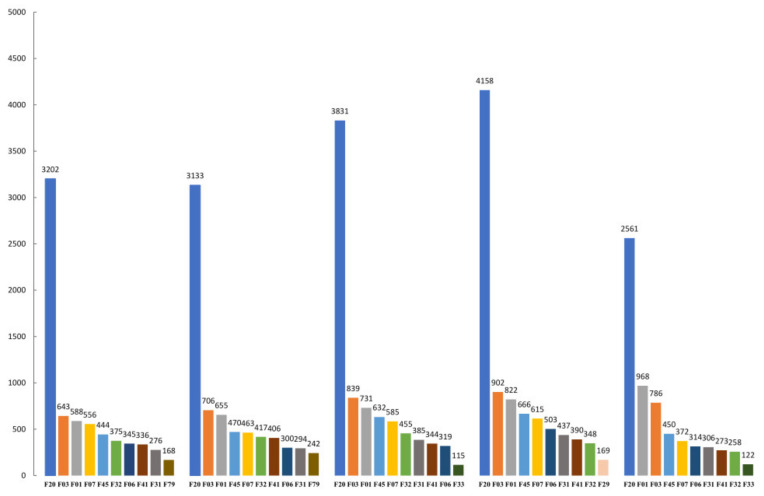
The top ten diseases for mental health service utilization at different years. Legend: The top ten diseases for mental health service utilization from January to July 2020 and the same period from 2016 to 2019. Note: F01, vascular dementia; F03, unspecified dementia; F06, other mental disorders due to brain damage and dysfunction and to physical disease; F07, personality and behavioral disorders due to brain disease, damage, and dysfunction; F20, schizophrenia; F31, bipolar affective disorder; F32, depressive episode; F33, recurrent depressive disorder; F41, other anxiety disorders; F45, somatoform disorders; F79, unspecified mental retardation.

**Figure 5 healthcare-10-01402-f005:**
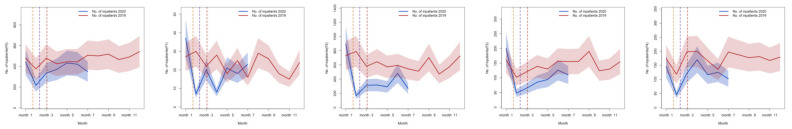
Utilization of the five major types of mental illness services from January 2019 to July 2020. Legend: The five major types of mental illness services utilization from January 2019 to July 2020. Note: F00–F09, organic, including symptomatic, mental disorders; F10–F19, mental and behavioral disorders due to psychoactive substance use; F20–F29, schizophrenia, schizotypal and delusional disorders; F30–F39, mood disorders; F40–F48, neurotic, stress-related, and somatoform disorders. The red line is the number of inpatients in 2019. The blue line is the number of inpatients from January to July 2020. The shades of blue and red represent the 95% CI of the regression line.

**Figure 6 healthcare-10-01402-f006:**
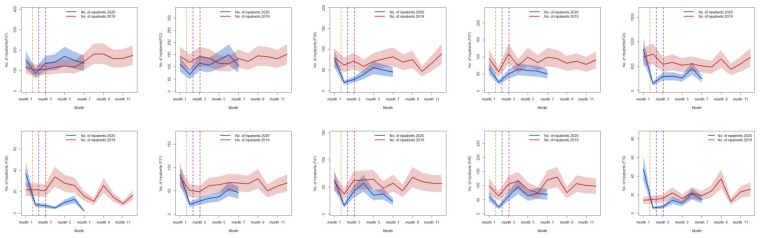
Utilization of ten mental illness services from January 2019 to July 2020. Legend: The ten diseases for mental health service utilization from January 2019 to July 2020. Note: F01, vascular dementia; F03, unspecified dementia; F06, other mental disorders due to brain damage and dysfunction and to physical disease; F07, personality and behavioral disorders due to brain disease, damage and dysfunction; F20, schizophrenia; F29, unspecified nonorganic psychosis; F31, bipolar affective disorder; F41, other anxiety disorders; F45, somatoform disorders; F79, unspecified mental retardation. The red line is the number of inpatients in 2019. The blue line is the number of inpatients from January to July 2020. The shades of blue and red represent the 95% CI of the regression line.

**Figure 7 healthcare-10-01402-f007:**
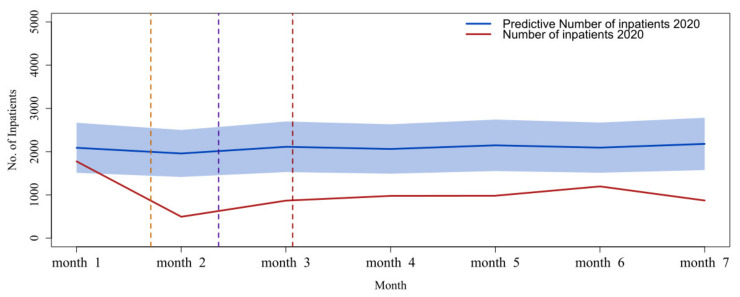
Comparison between observed inpatients and predicted inpatients from January to July 2020. The blue line is the ARIMA model estimate, and the blue-shaded region is the model 95% CI.

**Table 1 healthcare-10-01402-t001:** Characteristics of patients who visited the mental health inpatient service between 2016 to 2020.

Characteristics	2016	2017	2018	2019	2020
	N %	N %	N %	N %	N %
No. Patients	(20.0%)	(20.9%)	(23.5%)	(25.4%)	7166 (10.3%)
Gender					
Male	6804 (48.9%)	7177 (49.2%)	7966 (48.6%)	8683 (49.0%)	3431 (47.9%)
Female	7122 (51.1%)	7409 (50.8%)	8409 (51.4%)	9032 (51.0%)	3735 (52.1%)
Age, years					
≤44	4353 (31.3%)	4454 (30.5%)	5029 (30.7%)	5527 (31.2%)	2130 (29.7%)
45–64	3350 (24.1%)	3551 (24.3%)	4113 (25.1%)	4313 (24.3%)	1759 (24.5%)
65–74	3557 (25.5%)	3797 (26.0%)	4121 (25.2%)	4274 (24.1%)	1871 (26.1%)
≥75	2666 (19.1%)	2784 (19.1%)	3112 (19.0%)	3601 (20.3%)	1406 (19.6%)
Length of stay (days) M (P_25_, P_75_)	32 (13~81)	31 (12~76)	31 (13~71)	27 (11~60)	25 (11~59)
Diagnosis					
Organic, including symptomatic, mental disorders (F00–F09)	3914 (28.1%)	4192 (28.7%)	4938 (30.2%)	5661 (32.0%)	2563 (35.8%)
Mental and behavioral disorders due to psychoactive substance use (F10–F19)	203 (1.5%)	230 (1.6%)	231 (1.4%)	278 (1.6%)	134 (1.9%)
Schizophrenia, schizotypal and delusional disorders (F20–F29)	6033 (43.3%)	5984 (41.0%)	6979 (42.6%)	7439 (42.0%)	2729 (38.1%)
Mood disorders (F30–F39)	1326 (9.5%)	1426 (9.8%)	1681 (10.3%)	1720 (9.7%)	728 (10.2%)
Neurotic, stress-related and somatoform disorders (F40–F48)	1775 (12.7%)	2002 (13.7%)	2036 (12.4%)	2078 (11.7%)	825 (11.5%)
Behavior syndromes associated with physiological disturbances and physical factors (F50–F59)	139 (1.0%)	124 (0.9%)	132 (0.8%)	238 (1.3%)	61 (0.9%)
Disorders of adult personality and behavior (F60–F69)	26 (0.2%)	25 (0.2%)	31 (0.2%)	41 (0.2%)	10 (0.1%)
Mental retardation (F70–F79)	431 (3.1%)	503 (3.4%)	230 (1.4%)	173 (1.0%)	92 (1.3%)
Disorders of physiological development (F80–F89)	3 (0.0%)	3 (0.0%)	12 (0.1%)	12 (0.1%)	1 (0.0%)
Behavioral and emotional disorders that usually start in childhood and adolescence (F90–F99)	76 (0.5%)	97 (0.7%)	105 (0.6%)	75 (0.4%)	23 (0.3%)

**Table 2 healthcare-10-01402-t002:** The observed and predicted number of inpatients with mental health during the COVID-19 outbreak in Shanghai.

Date	January 2020	February 2020	March 2020	April 2020	May 2020	June 2020	July 2020
Inpatients							
Observed	1776	495	868	978	981	1197	871
Predicted	2090	1959	2113	2062	2147	2094	2180
95% CI for Predicted	1511 to 2669	1416 to 2502	1528 to 2699	1491 to 2634	1552 to 2742	1513 to 2674	1575 to 2784

Data are shown as number or 95% uncertainty interval.

## Data Availability

The data presented in this study are available in article or Appendix A.

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
