# Peer review of "Impact of the COVID-19 Pandemic on Utilization of Inpatient Mental Health Services in Shanghai, China"

_healthcare, 2022, doi:10.3390/healthcare10081402_

Round 1

Reviewer 1 Report

Thank you for the opportunity to review the manuscript. Overall, a current topic for a broader readership and further exploration of this topic is certainly important, especially to investigate how to be addressed to impact of COVID-19 pandemic on the utilization of the service in Shanghai by comparison with hospital-based health care records during the preceding 4 years!

A few questions / comments and suggestions:

In Line 211 to 299, the content of similar result with other studies as serious impact on mental health services throughout the world, suggest having more elaborate in other areas such as Asian and Australia.

In Line 238 to 239, more elaboration for this discussion. "During the COVID-19 pandemic and the Chinese New Year holiday. Any linkage with these events. Please provide more explanation and elaboration for other holidays such as Chinese National Day Holiday - Golden Week, etc.

In Line 255 to 257 - relevance to this study is not clear.

In Line 261 - relevance to this study is not clear, more supporting references.

In Line 267-269 - relevance to this study is not clear.

Author Response

Dear Reviewer,

Reviewer 2 Report

This article presents an analysis of the trends in the number of patients diagnosed of mental health issues, before and after the pandemic. The topic is highly relevant, although not novel, as shown in the literature review (there are similar studies in other countries).

Overall the study is concise and well presented. However, in some cases maybe it's too concise. In particular, I detected two main confusion points:

1- It is unclear to me how the division into the four stages was done. Was it done by the government, or by the authors? What is the rationale for such division?

2-Table 2 shows the predicted number of patients. Predicted by whom? How were those predictions made?

Author Response

Dear Reviewer,
